# Diabetic Ketoacidosis in Young Adults with Type 1 Diabetes: The Impact of the Ketogenic Diet—A Narrative Literature Review

**DOI:** 10.3390/diseases13100347

**Published:** 2025-10-17

**Authors:** Joanna Cielecka, Zuzanna Szkamruk, Maciej Walędziak, Anna Różańska-Walędziak

**Affiliations:** 1Medical Faculty, Collegium Medicum, Cardinal Stefan Wyszynski University in Warsaw, 01-938 Warsaw, Poland; 127466@student.uksw.edu.pl (J.C.); 128287@student.uksw.edu.pl (Z.S.); 2Department of General, Oncological, Metabolic and Thoracic Surgery, Military Institute of Medicine, 01-141 Warsaw, Poland; 3Department of Human Physiology and Pathophysiology, Faculty of Medicine, Collegium Medicum, Cardinal Stefan Wyszynski University in Warsaw, 01-938 Warsaw, Poland; aniaroza@tlen.pl

**Keywords:** diabetic ketoacidosis (DKA), type 1 diabetes (T1D), young adults, ketogenic diet, low-carbohydrate diet, euglycemic diabetic ketoacidosis (euDKA), nutritional ketosis, hyperglycemia

## Abstract

(1) Background: diabetic ketoacidosis (DKA) remains one of the most serious acute complications of type 1 diabetes, especially among young adults. At the same time the ketogenic diet, characterized by high fat and very low carbohydrate intake, is becoming increasingly popular, raising concerns about its appropriateness and safety for individuals with type 1 diabetes, (2) Methods: a literature review was conducted using MEDLINE and SCOPUS databases, complemented by additional searches in Embase, Cochrane Library, and Web of Science to ensure broad coverage of both international and European studies with the focus on keywords including “diabetic ketoacidosis”, “type 1 diabetes”, and “ketogenic diet”. The most relevant and up-to-date studies were selected to evaluate both risks and potential clinical applications of this diet in T1D in young adults, (3) Results and Conclusions: While nutritional ketosis under controlled conditions is typically safe, individuals with T1D, especially young adults, may be more vulnerable to DKA due to factors such as inconsistent insulin administration, lack of ketone monitoring, and lifestyle changes. Reports of euglycemic DKA further highlight the importance of regular ketone tracking, even when blood glucose appears within target ranges. Although low-carbohydrate diets may offer improved glycemic profiles, their use in young adults with T1D must be carefully evaluated, emphasizing individualized care plans, close metabolic monitoring, and comprehensive patient education. Ongoing research is essential to clarify whether ketogenic diet can be safely integrated into diabetes management in this population.

## 1. Introduction

### 1.1. Definition and Clinical Significance of the Topic

Diabetic ketoacidosis (DKA) is a potentially life-threatening metabolic disorder associated with type 1 diabetes mellitus (T1DM). It occurs due to an insufficient supply of insulin, leading to metabolic acidosis, excessive ketone body production and uncontrolled hyperglycemia. Despite progress in diabetes treatment DKA continues to be a major health concern, particularly among young adults. The ketogenic diet is becoming increasingly popular among youth with metabolic diseases. This diet is a very low-carbohydrate and high-fat diet [1]. It is characterized by reducing carbohydrates intake to 20–50 g/day, which indicates to about 4–10% of 2000 kcal/day diet. In the following section of our study, we will focus on analyzing its impact on the occurrence of diabetic ketoacidosis.

### 1.2. Pathophysiology, Physiology and Biochemistry of Diabetic Ketoacidosis

The primary trigger for DKA is an absolute or relative insulin deficiency resulting from autoimmune destruction of pancreatic β-cells in the islets of Langerhans. This insulin deficiency is accompanied by an increase in counter-regulatory hormones, such as glucagon, growth hormone, catecholamines and cortisol, which are elevated in response to physiological stress [2]. This leads to severe biochemical and clinical disturbances, such as significant hyperglycemia (typically above 11 mmol/L), metabolic acidosis (blood pH < 7.3 and/or HCO_3_^−^ concentration < 18 mmol/L), pronounced glucosuria with ketonuria, and dehydration (≥5%), which may be accompanied by vomiting and drowsiness [3]. Insulin deficiency, together with insulin resistance, leads to an imbalanced insulin-to-glucagon ratio. This triggers the activation of hormone-sensitive lipase, which catalyzes the breakdown of triglycerides stored in peripheral adipose tissue, releasing glycerol and long-chain fatty acids. Most of these fatty acids, bound to albumin, are transported through the splanchnic circulation and taken up by hepatocytes. Within liver cells, fatty acids undergo β-oxidation in the mitochondria after conjugation with coenzyme A (CoA), producing acetyl-CoA. Reduced insulin levels and increased glucagon activity further enhance the mitochondrial transport of acyl-CoA, a process mediated by carnitine palmitoyl transferase enzymes.

### 1.3. Purpose of the Study

The purpose of this study is to explore the potential impact of ketogenic dietary patterns on the occurrence and clinical presentation of diabetic ketoacidosis (DKA) in young adults with type 1 diabetes. With the ketogenic diet gaining traction as a glycemic control strategy, especially among younger individuals managing type 1 diabetes, concerns have arisen regarding its safety and potential to precipitate DKA. By examining clinical data and dietary habits, the study seeks to determine whether adherence to a ketogenic diet influences the risk profile for DKA, and to inform clinical guidelines on dietary recommendations for this vulnerable population.

## 2. Materials and Methods

The MEDLINE and SCOPUS scientific databases were searched for publications related to the ketogenic diet and diabetic ketoacidosis (DKA) in individuals with type 1 diabetes. To achieve a more comprehensive overview, supplementary searches were conducted in Embase, Cochrane Library, and Web of Science using Boolean operators (AND, OR) to combine relevant terms. Keywords used in the search included: “diabetic ketoacidosis (DKA)”, “type 1 diabetes (T1D)”, “young adults”, “ketogenic diet”, “low-carbohydrate diet”, “euglycemic diabetic ketoacidosis (euDKA)”, “nutritional ketosis”, “hyperglycemia”. Preference was given to original research articles and recent systematic reviews. Additional sources were identified through manual screening of reference lists in selected key publications. The most relevant and up-to-date studies offering clinical insight or highlighting areas of controversy were chosen for analysis. List of publication are presented in Table 1. All sources were last searched/consulted on 10 May 2025. The review was structured to reflect major thematic areas, including the pathophysiology and biochemical mechanisms of DKA, dietary ketosis versus pathological ketoacidosis, potential benefits and risks of ketogenic interventions in type 1 diabetes, and current approaches to monitoring, prevention, and management. Two reviewers independently screened titles, abstracts, and full texts according to predefined inclusion and exclusion criteria. Discrepancies were resolved by discussion and consensus. Data extraction was carried out using a standardized form to ensure consistency. A full description of the search strategy, including database-specific search strings, Boolean operators, limits, and date of the last search, is presented in Appendix A to ensure transparency and reproducibility. Due to the heterogeneity of study designs and the exploratory scope of this review, a formal risk of bias assessment (e.g., using ROBINS-I or Newcastle–Ottawa Scale) was not performed. However, the quality and reliability of each included study were assessed descriptively, based on study design, participant characteristics, and reporting completeness. This review was conducted in accordance with the Preferred Reporting Items for Systematic Reviews and Meta-Analyses (PRISMA 2020) guidelines. A quantitative synthesis (meta-analysis) was not performed due to the heterogeneity of study designs, participant characteristics, dietary interventions, and outcome measures among the included studies. Therefore, the results were summarized narratively to provide a comprehensive qualitative overview of the available evidence. The protocol of this review was not registered in PROSPERO or any other registry due to the exploratory nature and time constraints of the project. The authors acknowledge that prospective registration would have further enhanced methodological transparency. The certainty of evidence was not formally assessed using the GRADE approach due to the heterogeneity of study designs and outcomes, as well as the narrative nature of the review. Instead, the strength and consistency of findings were considered qualitatively when interpreting the overall evidence.

The study selection process is illustrated in the PRISMA 2020 flow diagram (Figure 1), which outlines the number of records identified, screened, excluded, and finally included in the review.

## 3. Results

### 3.1. Regulation of Acid-Base Balance in the Body

Maintaining acid-base homeostasis is a vital physiological process, especially during metabolic disturbances such as diabetic ketoacidosis. In healthy individuals, arterial blood pH is maintained within a narrow range of 7.35–7.45. In DKA, this balance is disrupted due to a combination of absolute or relative insulin deficiency and elevated levels of counter-regulatory hormones including glucagon, catecholamines, cortisol, and growth hormone [26]. This hormonal environment promotes enhanced lipolysis, resulting in the release of free fatty acids, which are converted in the liver via β-oxidation into ketone bodies—acetoacetate, β-hydroxybutyrate, and acetone. These are acidic in nature, and their accumulation in the bloodstream leads to high anion gap metabolic acidosis, the hallmark of DKA. To maintain pH within a range of survivability, the body activates several compensatory mechanisms. The bicarbonate buffer system acts as the first line of defense. Bicarbonate ions (HCO_3_^−^) bind to free hydrogen ions (H^+^), forming carbonic acid (H_2_CO_3_), which then dissociates into water and carbon dioxide (CO_2_). The CO_2_ is exhaled via the lungs, which temporarily stabilizes pH. As acidosis progresses and bicarbonate stores become depleted, respiratory compensation becomes increasingly important. Central chemoreceptors in the medulla oblongata detect lowered pH and stimulate Kussmaul respiration, a pattern of deep, rapid breathing aimed at expelling more CO_2_ and thereby elevating blood pH [26]. The renal system offers a slower but longer-term response by excreting hydrogen ions and reabsorbing bicarbonate. However, in the acute setting of DKA, this compensation is often limited due to volume depletion and reduced renal perfusion. Additionally, severe electrolyte imbalances, particularly in potassium and phosphate, can impair renal acid handling. If left untreated, the failure of these buffering and compensatory systems can lead to profound acidosis, dehydration, arrhythmias, cerebral edema, and ultimately death. Therefore, rapid correction of acid-base disturbances is a cornerstone of DKA management. Treatment involves intravenous insulin to suppress ketogenesis, fluid resuscitation to improve perfusion and kidney function, and cautious electrolyte replacement. As insulin suppresses lipolysis and ketone production, and as fluids restore perfusion, the acid-base balance is progressively normalized [27].

### 3.2. Triggering Factors and Risk Groups

The most frequent precipitating factor for diabetic ketoacidosis (DKA) in patients with type 1 diabetes mellitus (T1DM) is missed or inadequate insulin administration [4]. Insulin deficiency triggers the release of counter-regulatory hormones such as glucagon, catecholamines, cortisol, and growth hormone, further amplified by stress-related proinflammatory cytokines. These hormones stimulate lipolysis and proteolysis, increase hepatic glucose production, and enhance ketone formation. In the absence of insulin, glucose cannot be effectively metabolized via the citric acid cycle, leading to ketone accumulation, hyperglycemia, osmotic diuresis, and loss of electrolytes such as sodium, potassium, and chloride [28].

Infections are another common trigger of DKA, particularly bacterial infections caused by *Escherichia coli* and *Klebsiella pneumoniae* [29]. In older individuals, respiratory and urinary tract infections are most frequent and associated with increased mortality [5]. Prompt recognition and treatment of infections are therefore essential to prevent metabolic decompensation.

Vitamin D deficiency may also contribute to DKA susceptibility by impairing insulin secretion and sensitivity. It plays a role in pancreatic β-cell function and peripheral insulin responsiveness. Experimental studies have shown that vitamin D deficiency or receptor dysfunction reduces insulin secretion, while supplementation enhances calcium-mediated release [30]. Maintaining adequate vitamin D levels may thus support better metabolic stability in T1DM.

Thiamine (vitamin B1) is another micronutrient of importance, serving as a coenzyme for pyruvate dehydrogenase in aerobic glucose metabolism. Its deficiency, common in DKA, promotes anaerobic glycolysis and elevated lactate, potentially delaying recovery; supplementation may improve acid-base correction [31].

During DKA episodes, potassium balance is profoundly disturbed. Acidosis increases renal potassium loss due to competition with hydrogen ions in tubular reabsorption, leading to hypokalemia, which can cause serious cardiac complications if not properly corrected [32].

### 3.3. Clinical Picture

Patients with DKA often present with a spectrum of symptoms that reflect both metabolic and volume disturbances. The clinical onset of DKA is typically rapid, often developing over several hours to one or two days [33]. The initial symptoms reflect the consequences of unchecked hyperglycemia and ensuing volume depletion. Polyuria (increased urination) and polydipsia (excessive thirst) are among the earliest complaints, driven by osmotic diuresis due to elevated plasma glucose levels. As fluid losses accumulate, patients may develop dry mucous membranes, hypotension, and tachycardia. In more severe cases, hypovolemic shock can develop due to profound fluid loss. Dehydration signs, such as dry mucous membranes, reduced skin turgor, hypotension, and tachycardia, are typically evident on physical examination. Gastrointestinal symptoms such as nausea, vomiting, and diffuse abdominal pain are frequent and may sometimes mimic surgical emergencies [33,34]. In addition, patients commonly experience weakness, fatigue, and muscle cramps resulting from electrolyte disturbances, particularly hypokalemia and hypophosphatemia. Neurological impairment may range from mild drowsiness to deep coma and is often accompanied by Kussmaul respiration—characterized by deep, labored breathing—as well as a distinctive fruity odor on the breath caused by acetone exhalation.

### 3.4. Diagnostics

#### 3.4.1. Laboratory Evaluation

The diagnosis of DKA is confirmed through key biochemical criteria that assess glycemic control, acid-base status, ketone body production, and electrolyte balance.

#### 3.4.2. Blood Glucose Concentration

Although plasma glucose levels usually exceed 250 mg/dL (13.9 mmol/L), exceptions exist, notably in cases of euglycemic DKA, which may occur in patients on sodium-glucose cotransporter-2 (SGLT2) inhibitors or those with prolonged vomiting or poor oral intake [35,36].

#### 3.4.3. Acid-Base Balance

DKA is associated with metabolic acidosis, as evidenced by a blood pH below 7.30 and serum bicarbonate levels falling below 18 mmol/L. The severity of acidosis can be stratified as mild, moderate, or severe, based on these values. Venous blood gas analysis is often sufficient for evaluation and is less invasive than arterial sampling [37,38,39].

#### 3.4.4. Ketone Detection in Serum and Urine

Ketone bodies—mainly β-hydroxybutyrate and acetoacetate—accumulate as a consequence of enhanced fatty acid oxidation. Quantitative serum β-hydroxybutyrate levels above 3.0 mmol/L are highly suggestive of DKA. Urine dipstick tests detect acetoacetate, but may underestimate the degree of ketosis, especially early in the clinical course [40,41].

#### 3.4.5. Anion Gap and Electrolyte Disturbances

A raised anion gap metabolic acidosis is typical in DKA, calculated as:Anion gap = (Na^+^ + K^+^) − (Cl^−^ + HCO_3_^−^) 

Electrolyte imbalances—such as hyperkalemia (initially due to acidosis and insulin deficiency), hypokalemia (after treatment), and hyponatremia—are common. Monitoring and correcting potassium levels is critical before insulin administration, as insulin drives potassium into cells and may precipitate life-threatening hypokalemia if not addressed [42,43,44].

#### 3.4.6. Plasma Osmolality

Hyperosmolarity is a hallmark of severe DKA, contributing to neurological symptoms. Effective serum osmolality can be estimated using the formula:2 × [Na^+^] + [glucose]/18 + [BUN]/2.8

Elevated osmolality reflects the degree of dehydration and risk of cerebral edema, particularly in pediatric and young adult populations [26,45].

#### 3.4.7. Additional Testing

In the clinical assessment and management of diabetic ketoacidosis (DKA), especially in adolescent and young adult patients, a broad panel of laboratory tests is essential. These investigations not only help determine the severity of the condition but also guide the appropriate therapeutic interventions. Among the most relevant parameters are renal function markers, indicators of infection, and serum electrolyte levels. Dehydration may worsen renal perfusion. Renal Function Assessment (BUN and Creatinine). During episodes of DKA, significant dehydration due to osmotic diuresis may lead to a reduction in renal perfusion. This often results in elevated blood urea nitrogen (BUN) and creatinine levels, signaling possible acute kidney injury (AKI). A recent study by Sethi et al. (2020) observed that adolescents with DKA frequently presented with signs of transient renal dysfunction, which correlated with the degree of fluid loss and severity of acidosis (transient renal dysfunction denotes a temporary impairment in kidney function, often reversible after correction of dehydration and metabolic disturbances during DKA treatment) [6]. Markers of Infection (CRP and White Blood Cell Count). Since infections are a common precipitating factor for DKA, inflammatory markers like C-reactive protein (CRP) and white blood cell (WBC) count should be routinely assessed. It is important to recognize, however, that WBC count may be elevated in DKA even in the absence of infection, due to stress-induced leukocytosis. Therefore, interpreting these values requires a thorough clinical correlation, especially in younger patients [7].

#### 3.4.8. Electrolyte Disturbances (Phosphate and Magnesium)

Electrolyte imbalances are common in the early stages of DKA due to both urinary losses and insulin deficiency. Hypophosphatemia and hypomagnesemia, if not addressed, can contribute to complications like neuromuscular irritability, rhabdomyolysis, and arrhythmias [8,46]. Phosphate and magnesium levels should be closely monitored and supplemented as needed during treatment.

#### 3.4.9. Effects of Dehydration on Renal Perfusion

Dehydration remains a central feature of DKA, particularly in the pediatric and young adult population, where fluid losses can rapidly impair circulatory and renal function [9]. Delayed rehydration may aggravate renal hypoperfusion, increasing the risk of AKI. Early intravenous fluid therapy is, therefore, a cornerstone of treatment, aimed at restoring hemodynamic stability and protecting kidney function [10].

### 3.5. Complications and Consequences

Diabetic ketoacidosis (DKA) remains a life-threatening condition, particularly in young adults with type 1 diabetes, and its severity can be amplified by the metabolic shifts induced by a ketogenic diet.

#### 3.5.1. Cerebral Edema

One of the most dangerous acute complications is cerebral edema (CE). It refers to the abnormal buildup of fluid within and around brain cells, often linked to neurological conditions [47]. CE represents one of the most serious and life-threatening complications associated with DKA. While it is well-documented in children, affecting approximately 0.2–1% of pediatric DKA cases and being the primary cause of diabetes-related mortality in this population, it remains exceedingly uncommon among adults. This rarity is reflected in the limited number of case reports available over the past 20 years, highlighting just a few instances of CE in adult DKA patients. Among these cases, about half resulted in death, though those who survived generally recovered without lasting neurological impairment, Several risk factors have been linked to the development of CE, including a recent diagnosis of diabetes, younger age, experiencing DKA for the first time, more severe metabolic derangements at onset, the use of intravenous bicarbonate and slow correction of low sodium levels during treatment [48]. The onset of cerebral edema during treatment for diabetic ketoacidosis may by indicated by a series of evolving neurological and systemic symptoms. One of the earliest warning signs is a headache that either begins or intensifies after therapy has started. Changes in behavior, such as increasing irritability, confusion, unusual sleepiness, or even loss of bladder control, should raise concern. The appearance of localized neurological deficits, such as abnormalities in eye movement or pupil reactions, may also suggest progressing cerebral involvement. Recurrent vomiting following initial improvement can be another early clue. Cardiovascular signs, including a decreasing heart rate not explained by rest or rehydration, and an increase in blood pressure, may accompany the neurological symptoms. Breathing pattern irregularities, particularly if accompanied by low oxygen levels, are further cause for concern [49].

#### 3.5.2. Electrolyte Imbalances

Electrolytes are essential minerals found in the human body that are involved in various physiological functions. They help regulate fluid balance, maintain acid-base balance (pH), facilitate nerve signaling, support blood clotting, and enable muscle contraction [44]. Electrolyte disturbances are common and significant complications in young adults with type 1 diabetes experiencing diabetic ketoacidosis. Exogenous insulin administration can induce mild hypokalemia by promoting potassium (K^+^) movement into skeletal muscles and liver cells through increased Na^+^-K^+^-ATPase pump activity. Insulin-induced hypoglycemia may also contribute to this process by stimulating epinephrine secretion. In severe hyperglycemia, particularly during DKA, the majority of patients are sig potassium-depleted, with deficits ranging from 3 to 5 mEq/kg, and in some, exceeding 10 mEq/kg. Contributing factors to potassium depletion include vomiting, osmotic diuresis, ketoacid anion excretion, and cell K^+^ loss due to glycogen breakdown and protein catabolism. Although serum potassium levels may appear normal or elevated upon admission, this is often due to hyperosmolality and insulin deficiency, which cause water to move out of cells, leading to temporary rise in extracellular K^+^ concentration. The administration of insulin in this setting drives K^+^ back into cells, potentially causing severe hypokalemia, particularly in patients with initial normal or low potassium levels. It is critical to delay insulin therapy in hypokalemic patients until serum potassium exceeds 3.3 mEq/L to avoid life-threatening complications [43]. Effective management of electrolytes is essential in DKA to prevent severe complications, such as arrythmias, seizures, and cardiac arrest. The process of correcting electrolyte imbalances must be handled with caution, as rapid shifts can be dangerous. Therefore, it is crucial to monitor patients closely and develop personalized treatment strategies [42].

#### 3.5.3. Cardiovascular Complications

Acute cardiovascular events such as arrhythmias, myocardial infraction, and sudden cardiac death can occur during episodes of diabetic ketoacidosis, particularly due to underlying electrolyte disturbances. Among these, hypokalemia is the most frequent and poses the highest risk for life-threatening arrhythmias. While its severity can vary, even mild potassium deficits have been associated with abnormal electrocardiogram (ECG) findings, including ventricular ectopic beats. Additionally, changes in ST segment, such as depression, have been observed during DKA episodes and are considered early indicators of myocardial involvement. Another concerning finding is the prolongation of the corrected QT (QTc) interval, which is linked to an elevated risk of ventricular arrythmias, particularly Torsades de Pointes. Fortunately, QTc intervals have been shown to normalize with the solution of ketoacidosis, highlighting the reversible nature of some of these cardiac effects with appropriate management [11]. Troponin elevation observed during DKA can be misleading, as it often imitates signs of acute coronary syndrome, even when no true ischemic damage is present. Although such increases are typically associated with myocardial infraction, in the context of DKA, they may reflect myocardial stress caused by severe metabolic imbalance rather than direct coronary pathology. Contributing factors may include intense acidosis, elevated levels of stress hormones like catecholamines, and the harmful effects of free fatty acids on cardiac condition, especially in individuals with known cardiovascular risk factors. As traditional biomarkers and ECG changes may be nonspecific in this setting, further diagnostic evaluation, including imaging or cardiac consultation, may be necessary to accurately determine the etiology of troponin elevation [50].

#### 3.5.4. Mortality and Long-Term Consequences

Although diabetic ketoacidosis is a treatable condition, it still carries a significant risk of mortality, particularly when diagnosis or treatment is delayed. In young adults with type diabetes, DKA-related deaths are often linked to severe complications such as cerebral edema, cardiac arrhythmias, and multi-organ failure. Despite advances in treatment protocols, DKA remains a leading cause of morbidity and mortality in young adults with T1D. It is frequently observed at the time of diagnosis, and its incidence varies depending on healthcare access and socioeconomic factors. The most common trigger for DKA episodes is poor treatment adherence. While mortality during acute DKA episodes is relatively low, thanks to improved clinical protocols, recurrent DKA (rDKA) poses a serious long-term risk. Individuals experiencing rDKA often face challenges such as insulin delivery issues, ongoing hyperglycemia, substance use, and coexisting complications like diabetic neuropathy. This pattern is sometimes classified as part of “brittle diabetes” (the term “brittle diabetes” refers to a form of type 1 diabetes characterized by severe glycemic variability and recurrent episodes of hypoglycemia or ketoacidosis despite intensive insulin therapy). Recurrent episodes not only contribute to higher rates of hospital readmissions but also been associated with increased long-term mortality and psychological distress, including elevated risk of self-harm [12].

### 3.6. The Ketogenic Diet and Diabetic Ketoacidosis in Type 1 Diabetes—A Clinical Perspective for Adolescents and Young Adults

The ketogenic diet (KD), characterized by high fat, moderate protein, and very low carbohydrate intake, has gained popularity among individuals with type 1 diabetes (T1D), especially adolescents and young adults. It emphasizes unprocessed, nutrient-dense foods such as lean meats, fish rich in omega-3 fatty acids, healthy oils, avocados, nuts, seeds, and non-starchy vegetables, while discouraging processed low-carbohydrate products [51]. This dietary approach is often viewed as a means to improve glycemic control and metabolic stability.

Although several studies have shown potential benefits of KD—such as reduced glucose variability and lower insulin needs—its safety remains a major concern, particularly regarding diabetic ketoacidosis (DKA) [52,53]. Some reports suggest better glycemic outcomes and fewer hypoglycemic episodes, yet others warn that without appropriate insulin management, KD can precipitate severe complications, including DKA [13,54]. Ketogenesis, the process by which fatty acids are converted into ketone bodies, is the metabolic foundation of KD. In T1D, insufficient insulin disrupts this regulation, leading to uncontrolled ketone accumulation and risk of metabolic acidosis.

Euglycemic DKA (euDKA) is a rare but serious complication observed in T1D patients following KD. It is defined by ketosis and acidosis despite normal or mildly elevated glucose levels (<250 mg/dL) [55], which may delay diagnosis and treatment [36]. For instance, a 2024 case report described a 22-year-old woman who developed euDKA after starting KD without medical supervision, highlighting the need for proper insulin management and clinical monitoring [54]. In T1D, even slight reductions in insulin dosing may trigger DKA, as insulin normally suppresses excessive ketone production [56,57].

Distinguishing between nutritional and pathological ketosis is clinically important. Nutritional ketosis involves β-hydroxybutyrate (BHB) levels of 0.5–3.0 mmol/L with normal glucose and pH, whereas DKA presents with BHB > 3.0 mmol/L, pH < 7.3, and bicarbonate < 15 mmol/L, often with dehydration and electrolyte imbalance. Patients in nutritional ketosis remain stable, while those with DKA exhibit nausea, vomiting, tachypnea, and altered consciousness. Recognizing this difference—particularly in T1D patients following KD—is vital for early diagnosis and appropriate management.

The ketogenic diet inherently promotes ketone production; without sufficient insulin, this process becomes uncontrolled. Moreover, the absence of marked hyperglycemia can mask early DKA signs, leading to euglycemic presentations [58]. These risks are amplified in adolescents and young adults with fluctuating insulin needs due to growth, hormonal changes, and physical activity [59]. Therefore, careful supervision and education are crucial when KD is considered in this population.

Accurate ketone monitoring is essential for safety. Blood ketone meters provide faster and more reliable results than urine tests. Emerging continuous ketone monitoring (CKM) systems, integrated with continuous glucose monitoring (CGM), allow real-time detection of metabolic changes and early intervention [60]. Patient education on using these tools, recording results, and adjusting insulin under medical supervision improves safety and prevents DKA. Non-invasive ketone sensors and predictive algorithms using machine learning are also being developed to further enhance management [14].

The safety of KD in adolescents and young adults with T1D remains debated. While short-term improvements in glycemia have been observed, potential long-term effects on growth, nutrition, and cardiovascular health are concerning. Low-carbohydrate diets may impair linear growth and cause nutrient deficiencies [61], while high saturated fat intake could worsen lipid profiles and increase cardiovascular risk [62]. Current data on long-term ketosis in young populations remain limited.

Given these uncertainties, major health organizations recommend caution. The American Academy of Pediatrics does not support ketogenic or ultra-low-carbohydrate diets in children and adolescents with diabetes except under medical supervision. Similarly, the International Society for Pediatric and Adolescent Diabetes advocates individualized nutrition plans that ensure normal growth while maintaining glycemic targets [63].

In summary, while KD may provide short-term glycemic benefits in adolescents and young adults with T1D, its long-term effects on growth and cardiovascular health remain unclear. Individualized, medically supervised dietary approaches should be prioritized to balance metabolic control with overall development. (Reviewer 2)

### 3.7. Treatment and Therapeutic Strategies for Diabetic Ketoacidosis in Young Adults with Type 1 Diabetes

Effective management of DKA in young adults with type 1 diabetes necessitates a comprehensive approach that addresses insulin therapy, fluid resuscitation, electrolyte correction, vigilant monitoring, and preventive strategies to avert recurrence [64]. The essential steps in the acute management of diabetic ketoacidosis in young adults with type 1 diabetes are illustrated in Figure 2, starting with the patient’s presentation and progressing through fluid resuscitation, insulin therapy, and electrolyte correction.

#### 3.7.1. Intensive Insulin Therapy

In young adults with type 1 diabetes, intensive insulin therapy remains a central component of DKA treatment. After initial fluid resuscitation with 0.9% sodium chloride, insulin therapy is initiated once serum potassium levels are confirmed to be above 3.3 mEq/L to prevent hypokalemia-induced complications. The standard approach involves continuous intravenous insulin infusion, typically at a dose of 0.1 units/kg/hour. This method effectively suppresses ketone production, reduces hepatic glucose output, and enhances glucose uptake in peripheral tissues [15]. While some studies have explored the use of subcutaneous rapid-acting insulin analogs as an alternative for managing mild or moderate DKA in adults, particularly in settings with limited resources, evidence supporting their efficacy remains inconsistent. Most clinical guidelines continue to favor intravenous administration due to its predictability and rapid metabolic control. However, this approach required admission to high-dependency units or intensive care for continuous monitoring, which may not always be feasible in all healthcare settings. For young adults, selecting the appropriate insulin delivery method must be balance clinical severity, resource availability, and the need for close supervision [16].

#### 3.7.2. Fluid Management Strategies

Effective fluid resuscitation is essential in the treatment of diabetic ketoacidosis in young adults with type 1 diabetes. Traditionally, isotonic saline (0.9% sodium chloride) has been the initial fluid of choice. However, recent studies have explored the use of balanced electrolyte solutions (BES), such as Ringer’s lactate and Plasma-Lyte, as alternatives. Evidence suggests that while both fluid types are effective in managing DKA, BES may provide better outcomes in terms of acid-base balance. One analysis indicated that post-resuscitation bicarbonate levels were higher and chloride levels were lower in patients treated with BES, which may help prevent hyperchloremic metabolic acidosis, a complication sometimes seen with high-volume saline use [65]. Another study comparing BES and saline found no significant difference in the time to DKA resolution, but similarly reported improved bicarbonate and chloride profiles in the BES group [17]. Additionally, a meta-analysis found that BES use was linked to a slightly faster resolution of DKA and more favorable electrolyte parameters, without significant differences in mortality or insulin duration [18]. These findings highlight the importance of individualized fluid therapy in the management of DKA, taking into account the patient’s metabolic status and potential electrolyte disturbances, particularly in young adults. Although normal saline remains the standard of care, balanced electrolyte solutions (BES) may offer advantages in selected cases by reducing the risk of iatrogenic complications.

#### 3.7.3. Correction of Electrolyte Imbalances

Effective management of DKA in young adults with type 1 diabetes necessitates meticulous correction of electrolyte disturbances, particularly concerning potassium, sodium, and bicarbonate levels. In young adults with type 1 diabetes, effective fluid resuscitation plays a vital role in the early management of diabetic ketoacidosis. The primary goals of rehydration include restoring intravascular volume, improving tissue perfusion, and supporting renal function. Initial management begins with administering 10–20 mL/kg of isotonic saline or balanced crystalloid over 20–30 min, with a maximum of 1000 mL. In cases of significant dehydration or hypotension, additional boluses may be required. Following initial resuscitation, maintenance and deficit fluids are typically replaced over 36 h, often starting at double the normal maintenance rate. To prevent complications such as cerebral edema, careful monitoring of neurological status is essential. As blood glucose levels fall, dextrose is added to fluids to avoid hypoglycemia, aiming to maintain glucose between 7 and 11 mmol/L. Balanced crystalloids, such as Ringer’s lactate, may be preferred in some cases due to a lower risk of hyperchloremic acidosis. For patients with mild DKA who can tolerate oral intake, reduced intravenous fluid volumes or oral hydration may be appropriate [66]. Equally crucial is the correction of potassium imbalance, which should be initiated prior to insulin therapy if serum potassium is below 3.3 mmol/L, to prevent life-threatening cardiac arrhythmias. Sodium levels should be interpreted cautiously, particularly in the context of hyperglycemia, and corrected values should guide fluid selection. Bicarbonate therapy is generally not recommended except in cases of severe acidemia (pH < 6.9), where it may be considered to stabilize the cardiovascular system or during critical interventions such as intubation. Throughout treatment, continuous monitoring of electrolyte trends is necessary to adjust supplementation and prevent complications such as hypokalemia, hypernatremia, or paradoxical acidosis. Additionally, the pace of glucose correction should be controlled to reduce the risk of cerebral complications, especially in younger patients [67].

#### 3.7.4. Patient Monitoring and Hospitalization Criteria

In young adults with type 1 diabetes, reducing DKA incidence and improving long-term outcomes require both acute monitoring during hospitalization and the implementation of structured preventive strategies. Multidisciplinary care models that combine individualized insulin therapy, standardized diabetes self-management education, and strong community or outpatient support have demonstrated success in decreasing DKA-related hospital admissions and readmissions. Integrating more intensive insulin strategies including the use of insulin pumps or multiple daily injections, can contribute to better glycemic control when coupled with robust education programs. Structured diabetes education, particularly when delivered consistently in both inpatient and outpatient settings, empowers young adults to manage their condition more effectively. Programs aligned with recognized standards, such as the National Standards for Diabetes Self-Management Education and Support (DSMES), have been shown to reduce diabetes-related admissions by improving patient engagement, self-care, and adherence. Additionally, expending care beyond standard clinical visits, such as through home-based nursing, school clinics, or community-led initiatives, may offer additional support to high-risk individuals. Moreover, to ensure sustainable success, monitoring systems should not only track acute parameters like blood glucose and electrolyte levels during DKA episodes but also include ongoing follow-up and education to prevent recurrence. Such comprehensive, team-based interventions are critical to improving care and minimizing the burden of DKA in this vulnerable population [68]. Evidence from urban clinical settings emphasizes that consistent patient monitoring and engagement play a crucial role in reducing recurrent hospitalizations for DKA. Studies have shown that young adults frequently experience interruptions in care, with many lacking regular outpatient follow-up prior to hospital admission. In some populations, fewer than 5% of patients accessed diabetes specialty clinics before presenting with DKA. Additionally, poor adherence to prescribed insulin strategies remains a leading factor precipitating DKA episodes. This highlights the importance of early identification of high-risk individuals through structured outpatient monitoring and reinforced education. Incorporating targeted interventions to address medication nonadherence, such as personalized follow-up plans, psychosocial support, and increased accessibility to diabetes educators, can significantly decrease the likelihood of readmission. Such proactive strategies not only support better metabolic control but also reduce the long-term healthcare burden associated with DKA [19].

#### 3.7.5. Prevention Strategies

Preventing diabetic ketoacidosis in young adults with type 1 diabetes requires a multifaceted approach that includes education, early identification, and technological support. Raising awareness among patients, caregivers, and healthcare professionals about the early warning signs of hyperglycemia and DKA is essential to enable timely intervention and reduce the risk of severe complications. Structured educational programs that emphasize sick-day management, insulin adherence, and ketone monitoring can empower individuals to take proactive steps during periods of illness or stress. Integrating modern diabetes technologies, such as continuous glucose monitoring, insulin pump therapy, and ketone testing devices, into routine care can also improve metabolic control and help detect early signs of decompensation. Long-term prevention depends not only on acute clinical support but also on reinforcing self-management skills and ensuring that patients have access to consistent, personalized follow-up care. This comprehensive strategy can significantly reduce DKA episodes and improve quality of life [69].

### 3.8. Patient Education and Preventive Strategies

#### 3.8.1. The Importance of Self-Monitoring of Blood Glucose

Effective self-monitoring of blood glucose (SMBG) is an essential element of diabetes management, particularly in young adults with type 1 diabetes, as it directly influences glycemic control and helps prevent acute complications such as diabetic ketoacidosis. Recent clinical observations have reinforced the value of continuous glucose monitoring (CMG) in this population, demonstrating HbA1c outcomes regardless of insulin delivery method. Even among individuals using basic therapy approaches like multiple daily injections without carbohydrate counting, CMG use has been associated with significantly better glycemic outcomes. This highlights the critical role of glucose monitoring technologies in supporting patients at all stages of self-management proficiency. Regular use of CGM allows for timely adjustments to therapy and fosters greater engagement in self-care behaviors. Clinicians are encouraged to address barriers to CGM access and adherence, and emphasize its benefits during routine visits, ensuring that all young adults with T1D receive consistent opportunities to integrate this technology into their care plan [20].

#### 3.8.2. Ketone Monitoring and Education on Early Warning Signs

Recognizing early signs and symptoms of DKA and implementing appropriate sick day management are critical for preventing acute complications in young adults with type 1 diabetes. Brief educational interventions, when incorporated into routine clinic visits, have been shown to significantly improve patients’ awareness of DKA progression and encourage the use of emergency support resources, such as 24 h medical helplines. By enhancing understanding and promoting timely action, such strategies may help reduce unnecessary emergency department visits. Direct engagement of young adults, rather that solely their caregivers, appears especially effective in fostering greater self-reliance and responsiveness to early indicators of DKA [21].

#### 3.8.3. Educational Programs for Patients and Their Families

Addressing the psychological aspects of T1D is a key component of comprehensive educational programs for young adults and their families. Therapeutic interventions such as counseling, cognitive-behavioral therapy (CBT), and peer support have been shown to significantly improve emotional resilience and coping mechanisms in adolescents at risk of newly diagnosed with T1D. These services empower patients to manage anxiety and stress associated with health monitoring and disease progression, while also enhancing their capacity to adapt to the demands of diabetes self-care. Group-based support interventions, in particular, help reduce feelings of isolation by connecting individuals facing similar experiences, fostering a sense of community and mutual encouragement. CBT techniques, when tailored to the needs of adolescents, can reshape unhelpful thought patterns, reduce psychological distress, and promote healthier behavioral responses. Furthermore, involving family members in these interventions enhances communication, builds emotional support systems, and reinforces adherence to treatment plans. By integrating mental health resources into T1D educational initiatives, healthcare providers can offer a more holistic approach to disease management, one that recognizes the interplay between emotional well-being and effective self-care [70].

#### 3.8.4. The Importance of Modern Technologies

Modern technologies such as insulin pumps and continuous glucose monitoring (CGM) systems have become increasingly important in the management of type 1 diabetes particularly among children and adolescents. Over the past two decades, there has been a noticeable rise in the use of insulin pumps, particularly in patients under 15 years old, where it has become the standard treatment. This growing popularity can be attributed to the advancements in pump technology, including features like automatic insulin suspension during hypoglycemia and the availability of tubeless pumps, as well as increased accessibility due to healthcare systems reimbursements. Furthermore, the integration of insulin pumps with CGM systems has significantly improved glycemic control, reducing the frequency of severe hypoglycemia and DKA, while enhancing overall quality of like for patients. The combination of these technologies has revolutionized diabetes management, providing more precise control and greater flexibility for individuals with type 1 diabetes [22,23].

## 4. Discussion

Diabetic ketoacidosis (DKA) is a life-threatening complication of type 1 diabetes mellitus in young adults, characterized by insulin deficiency and elevated counter regulatory hormones, leading to hyperglycemia, ketonemia, and metabolic acidosis [56]. The ketogenic diet, characterized by low carbohydrate and high fat intake, induces a state of nutritional ketosis. While this metabolic state differs from DKA, individuals with T1D are at risk of transitioning from nutritional ketosis to DKA, especially if insulin therapy is inadequate. However, long-term adherence to a ketogenic diet under medical supervision has shown potential benefits in glycemic control without acute complications [71].

The primary cause of DKA is an absolute or relative deficiency of insulin, which can result from missed doses or inadequate insulin administration. Infections, such as pneumonia or urinary tract infections, are common precipitating factors, as they increase insulin requirements and can lead to DKA if not managed appropriately. Other contributors include metabolic stress from concurrent illnesses, dietary indiscretions, and deficiencies in essential vitamins like B1 and D. These vitamin deficiencies can impair glucose metabolism and insulin sensitivity, further exacerbating the risk of DKA. Understanding and addressing these risk factors are crucial in preventing the onset of DKA [24,72].

Management of DKA requires prompt recognition and structured therapeutic approach to reverse metabolic derangements and prevent complications. The key component of treatment includes fluid resuscitation, insulin therapy, and electrolyte replacement, especially potassium, which is often depleted despite normal or elevated serum levels. Traditionally, 0.9% saline has been the fluid of choice. However, concerns about hyperchloremic metabolic acidosis have prompted investigations into balanced electrolyte solutions (BES) as alternatives. Some recent studies suggest that balanced electrolyte solutions (BES) may support a more effective correction of acid-base disturbances in DKA compared to standard saline. BES have been associated with improved biochemical profiles, including better bicarbonate levels and lower chloride concentrations, which may aid in faster recovery and reduce the risk of complications [10,65].

Electrolyte imbalances, particularly involving potassium, are common in patients with DKA. These disturbances may present as either elevated or decreased serum potassium levels, depending on the stage of the condition and kidney function. Proper management of potassium levels is essential during treatment, as abnormalities can lead to serious complications, including arrhythmias [67].

The ketogenic diet has attracted significant interest for its potential benefits in improving blood sugar regulation and decreasing the need for insulin in people with type 1 diabetes (T1D) [54]. While this diet may offer short-term advantages, particularly for adults, its application in adolescents and young adults requires careful consideration due to the potential risks involved.

One of the most concerning risks is the development of diabetic ketoacidosis (DKA), which can include euglycemic diabetic ketoacidosis (euDKA). In cases of euDKA, blood glucose levels may stay within the normal range or rise only slightly, but the body still experiences a significant buildup of ketones, which leads to acidosis. This can make it difficult to identify the condition early. There have been reports of patients who developed euDKA after starting a ketogenic diet without adequate medical supervision, reinforcing the need for regular monitoring of both blood glucose and ketone levels. Proper adjustments to insulin doses are also essential to prevent this dangerous condition, especially in adolescents, who may not yet have the necessary skills to manage such a restrictive diet independently [73].

The long-term effects of the ketogenic diet on younger individuals, particularly in terms of safety and effectiveness, are not yet fully understood. Some studies have raised concerns about potential negative impacts on physical growth and development. Prolonged adherence to a diet that is low in carbohydrates and high in fats may impair growth, delay puberty, and affect bone health, leading to conditions like decreased bone mineralization [25]. Moreover, the restrictive nature of this diet can result in deficiencies of important nutrients, such as calcium, magnesium, and vitamins D and B, which are vital for optimal growth and metabolic function [61]. There is also concern about the possible long-term cardiovascular effects of a diet high in saturated fats [63].

In summary, although the ketogenic diet may provide certain benefits in terms of glycemic control for individuals with type 1 diabetes, its use among adolescents and young adults should be undertaken with great caution. The possible complications—including diabetic ketoacidosis, impaired growth, micronutrient deficiencies, and cardiovascular effects—necessitate close medical oversight. If such a diet is introduced, it should occur strictly under professional supervision, with frequent monitoring of glucose and ketone levels and appropriate insulin adjustments. More comprehensive studies are needed to clarify the long-term effects of the ketogenic diet on growth, development, and overall metabolic health in this group. A recent case report described improved glycemic stability and reduced insulin needs in a T1D patient following prolonged adherence to a ketogenic regimen; nonetheless, the authors highlighted the importance of individualized evaluation and careful follow-up to minimize associated risks [71].

Although the protocol of this review was not registered in PROSPERO or any other database, this limitation has been acknowledged, and the review was conducted in accordance with PRISMA 2020 guidelines to ensure methodological rigor and transparency.

## 5. Conclusions

It appears that nutritional ketosis, as achieved through a carefully controlled ketogenic diet, is fundamentally distinct from diabetic ketoacidosis (DKA), both underlying mechanisms and clinical consequences. It is plausible that, for some individuals with type 1 diabetes, such a diet may support metabolic outcomes, including better glucose control and reduced insulin dosage. However, these potential benefits seem to depend heavily on consistent monitoring, precise insulin administration, and individualized care. The risk of rapidly progressing to DKA, particularly in the setting of insufficient insulin, remains a serious concern. Additionally, reported cases of euglycemic DKA emphasize the complexity of managing ketosis in type 1 diabetes. Therefore, the ketogenic diet might be cautiously considered as an adjunct strategy in selected patients, but only within a framework of comprehensive medical supervision. Further clinical research is needed to explore the safety, long-term effects, and practical guidelines for implementing such dietary interventions without increasing the risk of life-threatening complications.

## Figures and Tables

**Figure 1 diseases-13-00347-f001:**
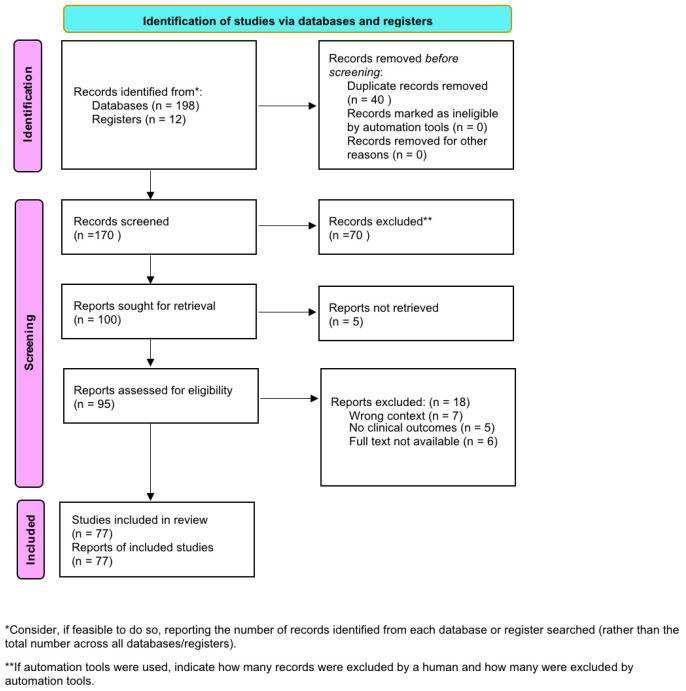
PRISMA 2020 flow diagram.

**Figure 2 diseases-13-00347-f002:**
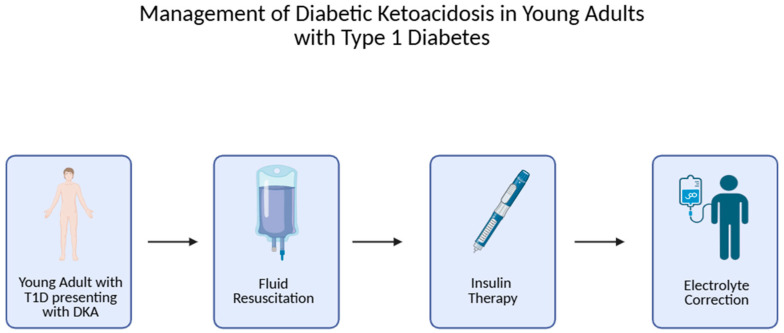
Management of diabetic ketoacidosis in young adults with type 1 diabetes: starting point—patient presenting with DKA; Block 1—fluid resuscitation with 0.9% NaCl and hemodynamic monitoring; Block 2—initiation of intravenous insulin therapy after fluids; Block 3—electrolyte correction with emphasis on potassium replacement.

**Table 1 diseases-13-00347-t001:** Publications related to the ketogenic diet and diabetic ketoacidosis.

Authors	Title	Journal	Type of Article	Number of Participants
Babiker A et al., 2022 [4]	Frequency and Risk Factors of Diabetic Ketoacidosis in a Specialized Children’s Hospital, Riyadh: A Cross-sectional Study	Oman Medical Journal	Original research	562
da Rosa Carlos Monteiro LE et al., 2022 [5]	Precipitating factors of diabetic ketoacidosis in type 1 diabetes patients at a tertiary hospital: a cross-sectional study with a two-time-period comparison.	Archives of endocrinology and Metabolism	Original research	97
Huang SK et al., 2020 [6]	Acute kidney injury is a common complication in children and adolescents hospitalized for diabetic ketoacidosis.	PLoS ONE	Original research	223
Tao LC et al., 2024 [7]	Inflammatory biomarkers predict higher risk of hyperglycemic crises but not outcomes in diabetic patients with COVID-19.	Frontiers in Endocrinology	Original research	124
Hasan RA, Hesen JZ et al., 2024 [8]	Serum Phosphorus and Hypophosphatemia During Therapy of Diabetic Ketoacidosis in Children: Single-Center, Retrospective Cohort 2016–2022.	Pediatric Critical Care Medicine	Original research	365
Trainor JL et al., 2023 [9]	Clinical and Laboratory Predictors of Dehydration Severity in Children With Diabetic Ketoacidosis.	Annals of Emergency Medicine	Original research	753
Szabó GV et al., 2024 [10]	Fluid resuscitation with balanced electrolyte solutions results in faster resolution of diabetic ketoacidosis than with 0.9% saline in adults—A systematic review and meta-analysis.	Diabetes/Metabolism Research and Reviews	Meta-analysis	1006
Aygün D, Aygün F et al., 2017 [11]	Electrocardiographic changes in children with diabetic ketoacidosis and ketosis.	Turkish Archives of Pediatrics	Original research	40
Santos SS et al., 2023 [12]	Increased risk of death following recurrent ketoacidosis admissions: a Brazilian cohort study of young adults with type 1 diabetes.	Diabetology and Metabolic Syndrome	Original research	231
Turton JL et al., 2023 [13]	Effects of a low-carbohydrate diet in adults with type 1 diabetes management: A single arm non-randomized clinical trial.	PLoS ONE	Clinical Trial	20
Song C et al., 2023 [14]	Point-of-Care Capillary Blood Ketone Measurements and the Prediction of Future Ketoacidosis Risk in Type 1 Diabetes.	Diabetes Care	Clinical Trial	484
Alshurtan KS et al., 2022 [15]	Efficacy and Safety of Intravenous Insulin in Treatment of Patient With Diabetic Ketoacidosis: A Systematic Review and Meta-Analysis.	Cureus	Meta-analysis	3258
Andrade-Castellanos CA et al., 2016 [16]	Subcutaneous rapid-acting insulin analogues for diabetic ketoacidosis.	Cochrane Library	Original research	201
Tamzil R, Yaacob N et al., 2023 [17]	Comparing the clinical effects of balanced electrolyte solutions versus normal saline in managing diabetic ketoacidosis: A systematic review and meta-analyses.	Turkish Journal of Emergency Medicine	Original research	595
Jozwiak M et al., 2024 [18]	Management of diabetic keto-acidosis in adult patients admitted to intensive care unit: an ESICM-endorsed international survey.	Critical Care	Original research	522
Wolf RA et al., 2019 [19]	Hospital admissions for hyperglycemic emergencies in young adults at an inner-city hospital.	Diabetes Research and Clinical Practice	Original research	273
Toschi E et al., 2021 [20]	Continuous Glucose Monitoring and Glycemic Control in Young Adults with Type 1 Diabetes: Benefit for Even the Simplest Insulin Administration Methods	Diabetes Technology and Therapeutics	Original research	888
Vitale RJ et al., 2018 [21]	An Effective Diabetic Ketoacidosis Prevention Intervention in Children With Type 1 Diabetes.	SAGE Open Nursing	Original research	76
Karges B et al., 2017 [22]	Association of Insulin Pump Therapy vs. Insulin Injection Therapy With Severe Hypoglycemia, Ketoacidosis, and Glycemic Control Among Children, Adolescents, and Young Adults With Type 1 Diabetes.	JAMA Network	Original research	30579
van den Boom L et al., 2019 [23]	Temporal Trends and Contemporary Use of Insulin Pump Therapy and Glucose Monitoring Among Children, Adolescents, and Adults With Type 1 Diabetes Between 1995 and 2017.	American Diabetes Association	Original research	96,547
Gonçalves SEAB et al., 2021 [24]	Association between thiamine deficiency and hyperlactatemia among critically ill patients with diabetes infected by SARS-CoV-2.	Journal of Diabetes	Original research	270
Levran N et al., 2023 [25]	The Impact of a Low-Carbohydrate Diet on Micronutrient Intake and Status in Adolescents with Type 1 Diabetes.	Nutrients	Clinical Trial	20

## Data Availability

No datasets were generated or analyzed during the current study.

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
