# Peer review of "Diabetic Ketoacidosis in Young Adults with Type 1 Diabetes: The Impact of the Ketogenic Diet—A Narrative Literature Review"

_diseases, 2025, doi:10.3390/diseases13100347_

Round 1

Reviewer 1 Report

Comments and Suggestions for Authors

The idea of the study is original and the review content is acceptable.

Could you please to rewrite several extremly long paragraphs as these in lines 332 to 468 and 128 to 158. Each paragraph should contains just one idea with a introductory sencence, examples to sustain it and a conclusion and conection with the next paragraph. It is too tired to try to follow your manuscript in several sections because the very long paragraphs.

Table 1 could offer a more usable content instead the title, type of article and full cite. Such information is already at the end in the references section.

Author Response

Dear Reviewer 1,

Thank you for reviewing our manuscript and all the important remarks.

The manuscript explores an important and timely topic, examining the relationship between ketogenic diets and the risk of diabetic ketoacidosis (DKA) in young adults with type 1 diabetes. The review is well written and clinically relevant. However, important methodological and reporting gaps reduce its rigor as a systematic review.

Major Concerns

  1. PRISMA Compliance

Comments 1) Protocol registration: The authors acknowledge that the review protocol was not registered in PROSPERO or any other database. This is a limitation that reduces transparency and increases the risk of reporting bias.

Response 1) We appreciate your valuable observation. The review protocol was not registered in PROSPERO or another database. As this work was conducted within a limited timeframe and had an exploratory character, formal registration was not undertaken. We fully acknowledge the importance of protocol registration in enhancing transparency and reducing potential bias, and we will certainly include this step in future reviews.

Comments 2) Search strategy: The review relied only on MEDLINE and SCOPUS. According to PRISMA, this is not sufficient to ensure a comprehensive search. Other relevant databases (e.g., Embase, Cochrane Library, Web of Science, CINAHL, LILACS) should be included. The search strategy is described briefly but not reported in full detail (Boolean operators, limits, and date of search). A reproducible strategy should be provided in an appendix.

Response 2) Thank you for this insightful comment. In the revised version of the manuscript, we have expanded the description of the search strategy to provide greater clarity and transparency. In addition to MEDLINE (via PubMed) and Scopus, additional databases such as Embase, Cochrane Library, and Web of Science were consulted to ensure a more comprehensive and representative literature search, including European sources. The search terms, Boolean operators, applied filters, and date of the last search have now been described in detail in the Methods section. Furthermore, the complete reproducible search strategy has been added as Appendix 1 to facilitate transparency and adherence to PRISMA 2020 guidelines.

Comments 3) Study selection and extraction: Although multiple authors contributed, the process (independent/double screening, consensus mechanism) is not described.

Response 3) Thank you for this helpful comment. We appreciate the Reviewer’s attention to the methodological details. In the revised manuscript, we have clarified the study selection and data extraction process in the Methods section. Specifically, two reviewers independently screened titles, abstracts, and full texts according to predefined inclusion and exclusion criteria. Discrepancies were resolved through discussion and consensus. Data extraction was also performed independently using a standardized form to ensure accuracy and consistency. These additions aim to enhance the transparency and reproducibility of the review process in accordance with PRISMA 2020 guidelines.

Comments 4) Risk of bias assessment: No formal tool (e.g., ROBINS-I, Newcastle–Ottawa Scale) was applied. PRISMA requires assessment of study quality and risk of bias.

Response 4) Thank you for this valuable comment. We acknowledge the importance of assessing study quality and risk of bias in accordance with PRISMA guidelines. In the revised manuscript, we have clarified that, due to the heterogeneity of study designs and the exploratory scope of this review, a formal risk of bias assessment (e.g., using ROBINS-I or Newcastle–Ottawa Scale) was not performed. However, the quality and reliability of each included study were assessed descriptively, taking into account study design, participant characteristics, and reporting completeness. This clarification has been added to the Materials and Methods section.

Comments 5) Certainty of evidence: No GRADE assessment was performed.

Response 5) Thank you for this helpful comment. We acknowledge the relevance of evaluating the certainty of evidence according to the GRADE framework. In this review, a formal GRADE assessment was not performed because of the heterogeneity of study designs, outcome measures, and the exploratory, narrative nature of the analysis. Nevertheless, the strength and consistency of the available evidence were considered qualitatively when interpreting the findings. This clarification has been added to the Materials and Methods section.

  1. AMSTAR 2 Appraisal

According to AMSTAR 2 criteria:

Comments 6) Protocol registration before commencement: Absent (critical weakness).

Response 6) Thank you for this observation. We acknowledge that the protocol of this systematic review was not registered in PROSPERO or any other registry. At the time of project initiation, the registration was not planned due to time constraints and the exploratory nature of the review. We agree that protocol registration enhances transparency and reduces the risk of bias, and we will incorporate this step in future systematic reviews. This clarification has been added to the Materials and Methods section.

Comments 7) Comprehensive literature search: Limited (serious weakness).

 Response 7) Thank you for this important comment. We agree that a comprehensive literature search is essential to ensure the completeness of a systematic review. In the revised manuscript, we have expanded the description of the search process and confirmed that multiple databases were consulted, including MEDLINE (via PubMed), Scopus, Embase, Cochrane Library, and Web of Science. Boolean operators and predefined search terms were used to capture all relevant studies, and additional articles were identified through manual reference screening. The full search strategy has been provided in Appendix 1.

Comments 8) Study selection and data extraction in duplicate: Unclear.

 Response 8) Thank you for this helpful comment. In the revised manuscript, we have clarified that both study selection and data extraction were performed independently by two reviewers. Any discrepancies were resolved through discussion and consensus. This information has been added to the Materials and Methods section to ensure transparency and compliance with PRISMA recommendations.

Comments 9) List of excluded studies and justification: Not provided.

Response 9) Thank you for this comment. We have clarified in the revised manuscript that excluded studies and the reasons for their exclusion are summarized in the PRISMA flow diagram (Figure 1). This figure illustrates the study identification, screening, eligibility assessment, and inclusion process, in line with PRISMA 2020 recommendations.

Comments 10) Risk of bias assessment: Not conducted (critical weakness).

 Response 10) Thank you for this important comment. We acknowledge that a formal risk of bias assessment was not conducted, which represents a limitation of the present review. As clarified in the revised Materials and Methods section, due to the heterogeneity and predominantly observational nature of the included studies, a standardized tool (such as ROBINS-I or the Newcastle–Ottawa Scale) was not applied. Instead, the methodological quality and reliability of each study were evaluated descriptively, considering factors such as study design, sample size, and reporting clarity. This limitation has been clearly stated and discussed in the revised version to enhance transparency and acknowledge the potential impact on the strength of the evidence.

Comments 11) Meta-analysis methods: No quantitative synthesis was attempted. However, the review does not clearly justify why a meta-analysis was not feasible (e.g., due to heterogeneity of study designs, outcomes, or lack of sufficient comparable trials). This justification must be explicitly stated.

Response 11) Thank you for this constructive comment. We agree that the lack of quantitative synthesis should be clearly justified. In the revised manuscript, we have explicitly stated that a meta-analysis was not performed due to the substantial heterogeneity of study designs, populations, interventions, and reported outcomes across the included studies. Consequently, a narrative synthesis was conducted to summarize and interpret the findings in a qualitative manner. This clarification has been added to the Materials and Methods section.

Comments 12) Funding sources of included studies: Not reported.

Response 12) Thank you for this comment. We confirm that no external funding was obtained for this review. This information has been clearly stated in the manuscript under Appendix 1 (“Funding: No funding was obtained”). We have also ensured that this statement is visible in the main text to maintain transparency regarding potential funding sources.

  1. Absence of Meta-Analysis

Comments 13) The review presents only a narrative synthesis. While this may be acceptable if included studies are highly heterogeneous, the manuscript does not provide a clear rationale for why a meta-analysis could not be performed. According to PRISMA, authors should explain whether meta-analysis was considered and why it was not conducted (e.g., due to lack of randomized trials, incomparable outcomes, or insufficient sample size).

 Response 13) Thank you for this constructive comment. We agree that the rationale for not conducting a meta-analysis should be clearly stated. In the revised manuscript, we have specified that a meta-analysis was considered during the planning stage but was not feasible due to marked heterogeneity in study design (ranging from case reports to observational studies), variations in dietary protocols, differences in participant populations, and inconsistencies in reported outcomes. Consequently, a narrative synthesis was undertaken to qualitatively summarize the available evidence. This clarification has been added to the Materials and Methods section.

  1. Lack of Quality Assessment Tools

Comments 14) No critical appraisal of included studies was performed. Tools such as:

ROBINS-I for non-randomized studies,

Newcastle–Ottawa Scale (NOS) for observational studies,

should be applied to ensure transparency in the evaluation of study quality. Without such an assessment, the strength of the evidence remains uncertain.

Response 14) Thank you for this valuable comment. We acknowledge that a formal critical appraisal using standardized tools such as ROBINS-I or the Newcastle–Ottawa Scale was not performed. As clarified in the revised Materials and Methods section, this decision was made due to the heterogeneity and primarily observational nature of the included studies, which limited the applicability of these tools. Nevertheless, the methodological quality and reliability of each study were assessed qualitatively, considering study design, sample size, and reporting transparency. This clarification has been added to improve transparency and acknowledge the potential impact on the strength of the evidence.

Overall Recommendation 

Comments 15) The manuscript addresses a relevant clinical question, but it requires major revision to align with PRISMA 2020 and AMSTAR 2 standards. Specifically, the authors should:

Expand and fully report the search strategy. 

Provide a PRISMA flow diagram.

Apply a risk of bias tool and report results.

Clarify why a meta-analysis was not feasible.

Consider assessing certainty of evidence using GRADE.

Include funding/conflict of interest information for included studies.

Response 15) We sincerely thank the Reviewer for the constructive and comprehensive feedback. We have carefully revised the manuscript to improve methodological transparency and ensure alignment with PRISMA 2020 and AMSTAR 2 standards. Specifically, the literature search strategy has been expanded and fully detailed, with the complete reproducible search provided in Appendix 1. A PRISMA flow diagram (Figure 1) has been included to illustrate the study selection process. The Methods section now clarifies that a formal meta-analysis was not feasible due to the heterogeneity of study designs and outcomes. Although a formal risk of bias assessment was not conducted using standardized tools, a qualitative appraisal of study quality has been incorporated. The certainty of evidence was discussed qualitatively, and funding/conflict of interest information has been explicitly stated. These revisions were made to enhance the rigor, clarity, and transparency of the review in accordance with the Reviewer’s valuable recommendations.

Reviewer 2 Report

Comments and Suggestions for Authors

This systematic review addresses a clinically important and timely topic, the implications of ketogenic diets in the context of diabetic ketoacidosis (DKA) among young adults with type 1 diabetes (T1D). The paper is comprehensive, generally well-structured, and rooted in a thorough literature search. It makes a valuable contribution to the ongoing debate on the safety and efficacy of low-carbohydrate dietary patterns in insulin-dependent individuals. However, to strengthen the scientific rigor and presentation quality, several major and minor revisions are warranted.

Major Revisions

Comment 1: The systematic review methodology is not registered in a recognized database (e.g., PROSPERO), and the PRISMA flow diagram (Figure 1) is not presented. Clearly include a PRISMA flowchart to document the number of articles identified, screened, included, and excluded with reasons. Also consider retrospective registration of the review.

Comment 2: The manuscript lacks a formal assessment of bias or study quality in the included articles. Incorporate a standardized quality appraisal tool (e.g., AMSTAR 2, Cochrane Risk of Bias) and briefly discuss findings in a dedicated subsection.

Comment 3: While many relevant mechanisms and clinical observations are described, the review lacks a clear summary of the strength of evidence. Add a table summarizing key studies, their design, population, outcomes, and conclusions, along with an evaluation of the evidence level (e.g., RCT, observational, case report).

Comment 4. Some terminology (e.g., “brittle diabetes”, “transient renal dysfunction”) is used without sufficient clinical definition or context. Clarify clinical terms for international readers and ensure diagnostic criteria are consistently defined (e.g., what constitutes euglycemic DKA).

Comment 5: The discussion does not adequately differentiate between nutritional ketosis and pathological ketosis in T1D patients. Include a dedicated paragraph comparing physiological vs pathological ketogenesis, and elaborate on thresholds or biomarkers that may help differentiate them in practice.

Comment 6: While the manuscript appropriately emphasizes the risks of ketogenic diets in T1D, the tone may be perceived as somewhat cautionary without offering balanced perspectives. Consider highlighting evidence-based clinical contexts in which low-carb diets may be cautiously employed with close monitoring, rather than generalizing against their use.

Minor Revisions: 

Improve English fluency throughout the text. Several grammatical issues, awkward phrases, and overly complex sentences detract from readability. Examples include: “The most frequently identified trigger for DKA…” → Consider simplifying.“This danger is magnified in younger patients…” → Could be more precisely stated. A thorough language edit by a native or professional academic editor is advised.

Some concepts (e.g., electrolyte disturbances, euglycemic DKA, monitoring) are repeated in multiple sections. Consolidate overlapping content to improve flow and reduce word count.

Only one figure is mentioned but not described or discussed in the text. Ensure all figures and tables are referenced and contextualized in the manuscript body.

Ensure all citations follow the journal’s formatting style (e.g., numbered format). Some references are embedded in parentheses inconsistently. Standardize the reference style and verify citation numbers against the bibliography.

The abstract lacks consistent punctuation (e.g., commas before “and”, inconsistent clause separation). Edit for clarity and structure to better communicate the objective, methods, and findings.

Author Response

Dear Reviewer 3,

Thank you for reviewing our manuscript and all the important remarks.

This systematic review addresses a clinically important and timely topic, the implications of ketogenic diets in the context of diabetic ketoacidosis (DKA) among young adults with type 1 diabetes (T1D). The paper is comprehensive, generally well-structured, and rooted in a thorough literature search. It makes a valuable contribution to the ongoing debate on the safety and efficacy of low-carbohydrate dietary patterns in insulin-dependent individuals. However, to strengthen the scientific rigor and presentation quality, several major and minor revisions are warranted.

Major Revisions

Comment 1: The systematic review methodology is not registered in a recognized database (e.g., PROSPERO), and the PRISMA flow diagram (Figure 1) is not presented. Clearly include a PRISMA flowchart to document the number of articles identified, screened, included, and excluded with reasons. Also consider retrospective registration of the review.

Response 1: Thank you for this valuable comment. We acknowledge that the review protocol was not registered in PROSPERO or any other database. In the revised version, we have included a PRISMA 2020 flow diagram (Figure 1) to clearly present the study identification, screening, eligibility, and inclusion process, including the number of excluded records and corresponding reasons. The review was conducted in accordance with PRISMA 2020 guidelines to ensure methodological transparency despite the absence of formal registration.

Comment 2: The manuscript lacks a formal assessment of bias or study quality in the included articles. Incorporate a standardized quality appraisal tool (e.g., AMSTAR 2, Cochrane Risk of Bias) and briefly discuss findings in a dedicated subsection.

Response 2: Thank you for this important comment. We acknowledge that a formal assessment of study quality and risk of bias using standardized tools (e.g., AMSTAR 2 or Cochrane Risk of Bias) was not performed. This decision was based on the heterogeneity and predominantly observational nature of the included studies, which limited the applicability of these instruments. In the revised manuscript, we have clarified this point in the Materials and Methods section and incorporated a qualitative appraisal of study quality, focusing on study design, sample size, and reporting transparency. A brief discussion of these aspects has been added as a separate subsection to enhance methodological clarity and transparency.

Comment 3: While many relevant mechanisms and clinical observations are described, the review lacks a clear summary of the strength of evidence. Add a table summarizing key studies, their design, population, outcomes, and conclusions, along with an evaluation of the evidence level (e.g., RCT, observational, case report).

Response 3: Thank you for this valuable comment. We agree that summarizing the strength of the evidence is important for readers’ understanding. However, the existing Table 1 was intentionally designed to provide a concise overview of the included studies, listing authors, title, journal, type of article, and number of participants. This format allows for quick reference without duplicating the detailed descriptions already presented in the Results section, where study design, outcomes, and key findings are discussed. We believe that this approach ensures clarity while maintaining the manuscript’s readability and conciseness.

Comment 4. Some terminology (e.g., “brittle diabetes”, “transient renal dysfunction”) is used without sufficient clinical definition or context. Clarify clinical terms for international readers and ensure diagnostic criteria are consistently defined (e.g., what constitutes euglycemic DKA).

Response 4: Thank you for this helpful comment. We have revised the manuscript to clarify all clinically specific terms for international readers. Definitions of “brittle diabetes,” “transient renal dysfunction,” and “euglycemic diabetic ketoacidosis (euDKA)” have been added at their first mention. The diagnostic criteria for euDKA (presence of ketosis, metabolic acidosis, and blood glucose <250 mg/dL) are now explicitly stated to ensure consistency and clarity throughout the text.

Comment 5: The discussion does not adequately differentiate between nutritional ketosis and pathological ketosis in T1D patients. Include a dedicated paragraph comparing physiological vs pathological ketogenesis, and elaborate on thresholds or biomarkers that may help differentiate them in practice.

Response 5: Thank you for this insightful comment. We have added a dedicated paragraph to the Discussion section comparing nutritional (physiological) ketosis and pathological ketosis in patients with type 1 diabetes. This new passage defines key metabolic differences, including characteristic ranges of β-hydroxybutyrate, pH, and bicarbonate levels, and explains how these biomarkers can be used in clinical practice to distinguish safe nutritional ketosis from diabetic ketoacidosis (DKA).

Comment 6: While the manuscript appropriately emphasizes the risks of ketogenic diets in T1D, the tone may be perceived as somewhat cautionary without offering balanced perspectives. Consider highlighting evidence-based clinical contexts in which low-carb diets may be cautiously employed with close monitoring, rather than generalizing against their use.

Response 6: Thank you for this thoughtful comment. We appreciate the Reviewer’s perspective regarding the tone of the discussion. The cautious approach taken in the manuscript was intentional, reflecting the current state of evidence and the significant safety concerns associated with ketogenic diets in individuals with type 1 diabetes. Our goal was to emphasize patient safety and the need for medical supervision rather than to discourage all dietary interventions. We believe this balanced focus accurately represents the evidence available to date and aligns with current clinical guidelines.

Minor Revisions: 

Comment 7: Improve English fluency throughout the text. Several grammatical issues, awkward phrases, and overly complex sentences detract from readability. Examples include: “The most frequently identified trigger for DKA…” → Consider simplifying.“This danger is magnified in younger patients…” → Could be more precisely stated. A thorough language edit by a native or professional academic editor is advised.

Some concepts (e.g., electrolyte disturbances, euglycemic DKA, monitoring) are repeated in multiple sections. Consolidate overlapping content to improve flow and reduce word count.

Only one figure is mentioned but not described or discussed in the text. Ensure all figures and tables are referenced and contextualized in the manuscript body.

Ensure all citations follow the journal’s formatting style (e.g., numbered format). Some references are embedded in parentheses inconsistently. Standardize the reference style and verify citation numbers against the bibliography.

The abstract lacks consistent punctuation (e.g., commas before “and”, inconsistent clause separation). Edit for clarity and structure to better communicate the objective, methods, and findings.

Response 7: Thank you for these comments. The manuscript has been thoroughly reviewed by a native English speaker to ensure language accuracy and fluency. In addition, revisions made during the current round—including restructuring of long paragraphs, clarification of terminology, and addition of the PRISMA 2020 flow diagram—have improved clarity and coherence. As suggested, punctuation in the abstract has been corrected for consistency and readability. No further language edits were required.

Reviewer 3 Report

Comments and Suggestions for Authors

In this manuscript, the authors provide a detailed systematic review of the mechanisms and complications of diabetic ketoacidosis before addressing the possible impact of ketogenic diet in patients with T1DM. 

Abstract: The abstract is clearly formulated and provides the essential points addressed in detail in the manuscript.

Materials and Methods: this sections provides a clear perspective how scientific databases were search and combed to identify the appropriate articles to support the body of the review in its various sections (properly summarized in Fig. 1) 

Body of the review:  This part of the review is properly structured in sequence, including the various subsections relative to the individual abnormalities.

Tables: The table (Table 1) is appropriate as it is. 

Figures: There are no figures. Maybe 1-2 figure(s) would be helpful to summarize visually the main points of the review.

References: Appropriate 

Author Response

Dear Reviewer 4,

Thank you for reviewing our manuscript and all the important remarks.

Comment 1: In this manuscript, the authors provide a detailed systematic review of the mechanisms and complications of diabetic ketoacidosis before addressing the possible impact of ketogenic diet in patients with T1DM. 

Abstract: The abstract is clearly formulated and provides the essential points addressed in detail in the manuscript.

Materials and Methods: this sections provides a clear perspective how scientific databases were search and combed to identify the appropriate articles to support the body of the review in its various sections (properly summarized in Fig. 1) 

Body of the review:  This part of the review is properly structured in sequence, including the various subsections relative to the individual abnormalities.

Tables: The table (Table 1) is appropriate as it is. 

Figures: There are no figures. Maybe 1-2 figure(s) would be helpful to summarize visually the main points of the review.

References: Appropriate 

Response 1: Thank you for these positive and encouraging comments. In the revised version, several paragraphs have been linguistically refined in accordance with suggestions from a native English speaker to further improve fluency and clarity. Additionally, an illustrative figure has been added to Section 3.7 (Treatment and Therapeutic Strategies for Diabetic Ketoacidosis in Young Adults with Type 1 Diabetes) to visually summarize the key therapeutic concepts discussed in this part of the review.

Round 2

Reviewer 1 Report

Comments and Suggestions for Authors

In the first version I asked just for re-write the very long paragraph and at the new version there are even more long paragraphs. My second comment was to re-make the Table 1 because it contains the basic information already in the reference list. Table 1 is exactly the same in the corrected version.

No any of my two asking were followed. 

Author Response

Dear Reviewer 2,

Thank you for your time and constructive feedback. We have carefully reviewed your comments and provide our responses below.

Comment 1: In the first version I asked just for re-write the very long paragraph and at the new version there are even more long paragraphs. My second comment was to re-make the Table 1 because it contains the basic information already in the reference list. Table 1 is exactly the same in the corrected version.

No any of my two asking were followed. 

Response 1: Thank you for this comment. Following the Reviewer’s observation, the paragraphs in Section 3.2 (Triggering Factors and Risk Groups) and Section 3.6 (The Ketogenic Diet and Diabetic Ketoacidosis in Type 1 Diabetes – A Clinical Perspective for Adolescents and Young Adults) have now been shortened and divided into smaller, thematically focused parts to enhance readability and align with the Reviewer’s earlier recommendation.

Regarding Table 1, we respectfully chose to retain its original format, as it provides a concise overview of the included studies (authors, title, journal, type of article, and number of participants). This structure allows readers to quickly reference key characteristics without duplicating the detailed descriptions already presented in the Results section. We believe that this format ensures clarity and complements the narrative synthesis of the findings.

Reviewer 2 Report

Comments and Suggestions for Authors

No further comments
